# Evaluation of Whole-Cell and Acellular Pertussis Vaccines in the Context of Long-Term Herd Immunity

**DOI:** 10.3390/vaccines11010001

**Published:** 2022-12-20

**Authors:** Ewa Szwejser-Zawislak, Mieszko M. Wilk, Piotr Piszczek, Justyna Krawczyk, Daria Wilczyńska, Daniela Hozbor

**Affiliations:** 1Institute of Biotechnology of Serums and Vaccines Biomed, Al. Sosnowa 8, 30-224 Krakow, Poland; 2Department of Immunology, Faculty of Biochemistry, Biophysics and Biotechnology, Jagiellonian University, 30-387 Krakow, Poland; 3VacSal Laboratory, Institute of Biotechnology and Molecular Biology, Faculty of Sciences, National University of La Plata (UNLP), National Council for Scientific and Technical Research (CONICET), La Plata 1900, Argentina

**Keywords:** pertussis, *Bordetella pertussis*, acellular pertussis vaccine, whole cell pertussis vaccine, DTaP, Tdap, DTwP

## Abstract

After the pertussis vaccine had been introduced in the 1940s and was shown to be very successful in reducing the morbidity and mortality associated with the disease, the possibility of improving both vaccine composition and vaccination schedules has become the subject of continuous interest. As a result, we are witnessing a considerable heterogeneity in pertussis vaccination policies, which remains beyond universal consensus. Many pertussis-related deaths still occur in low- and middle-income countries; however, these deaths are attributable to gaps in vaccination coverage and limited access to healthcare in these countries, rather than to the poor efficacy of the first generation of pertussis vaccine consisting in inactivated and detoxified whole cell pathogen (wP). In many, particularly high-income countries, a switch was made in the 1990s to the use of acellular pertussis (aP) vaccine, to reduce the rate of post-vaccination adverse events and thereby achieve a higher percentage of children vaccinated. However the epidemiological data collected over the past few decades, even in those high-income countries, show an increase in pertussis prevalence and morbidity rates, triggering a wide-ranging debate on the causes of pertussis resurgence and the effectiveness of current pertussis prevention strategies, as well as on the efficacy of available pertussis vaccines and immunization schedules. The current article presents a systematic review of scientific reports on the evaluation of the use of whole-cell and acellular pertussis vaccines, in the context of long-term immunity and vaccines efficacy.

## 1. Introduction

Pertussis, also known as whooping cough, is a highly contagious respiratory disease mainly caused by gram-negative bacteria named *Bordetella pertussis*. This is a disease that causes uncontrollable violent coughing, most commonly affects infants and young children, and can be fatal, especially in babies less than 1 year of age. However, it is important to note that pertussis can affect people of any age [1]. Several underlying medical conditions, such as asthma and chronic obstructive pulmonary disease, were reported as risk factors for pertussis [2,3,4]. The best way to prevent pertussis is to get vaccinated. The types of pertussis vaccines and schedules used in the national immunization programs have evolved since the development of the first generation of pertussis vaccine consisting of a suspension of detoxified and heat-killed bacteria (whole-cell pertussis, wP) [5,6,7,8,9]. In the late 1940s, Pearl Kendrick, Grace Eldering, and Loney Gordon developed a combined vaccine containing diphtheria and tetanus toxoids, as well as the whole cell pertussis component to obtain the DTP vaccine, which was widely adopted [10]. The Committee on Infectious Diseases of the American Academy of Pediatrics suggested in 1944, and recommended in 1947, the routine use of this vaccine, and the recommendation was later adopted in other countries. Vaccination coverage improved when the Expanded Program on Immunization (EPI) was established in 1974. By the early 1980s, mass vaccination against pertussis drastically reduced the morbidity and mortality associated with the disease. Shortly afterwards, as the risks from pertussis decreased markedly, attention shifted from the risk of the disease to the fear of vaccine side effects. Doubts began to arise about the safety of wP vaccines, which led to a decrease in the populational acceptance of this type of formulation; in some countries, its use was completely rejected [11]. For instance, in the United Kingdom and the US, concerns about the safety of this vaccine were widely reported in the popular press [12] and television documentary programs, such as “DTP: Vaccine Roulette” (WRY-TV, 1982), resulting in decreasing coverage [13]. In Japan, the government suspended pertussis vaccinations in 1975, due to the publicity about two suspected vaccine-related deaths in children. Two years later, the suspension of the vaccination program resulted in an increase in pertussis cases and 40 deaths [14]. The reactogenicity of wP was extensively evaluated in DTP, and the pertussis component proved to be the main factor responsible for the toxicity of these combined vaccines. Reported adverse reactions ranged from local reactions (redness, swelling, and pain at the injection site) to systemic reactions (fever, persistent crying, and in rare cases, encephalopathy). Safety concerns ultimately led to the development of the first acellular pertussis vaccine (aP), namely the Japanese vaccine developed by Sato et al., containing purified antigens from *B. pertussis* mixed with Alum as an adjuvant [15]. This aP vaccine had an improved safety profile over the Japanese whole-cell vaccine, while demonstrating comparable efficacy and was, therefore, implemented for use in two-year-old children [16]. The effectiveness of this vaccine was evidenced by the steady decline in the incidence of the disease in Japan [17]. The first randomized controlled study on aP vaccines was conducted in Sweden, where two Japanese vaccines, one containing formaldehyde-inactivated PT and FHA and the other only PT toxoid, were investigated and compared to a placebo control [18]. The study confirmed the improved safety profile of aP over wP vaccines and demonstrated comparable culture-confirmed pertussis prevention efficacies of about 80% after two doses. Similar methods for the production of acellular vaccines were adopted in other countries, usually with additional antigens, such as pertactin and serotype 2 and 3 fimbriae, being combined with diphtheria and tetanus toxoids (DTaP) [19,20]. Preparations containing up to five components were developed, and several efficacy trials clearly demonstrated that the aP vaccines were able to confer short-term protection that was comparable to that of the most effective wP vaccines, with fewer local and systemic reactions [21]. By the late 1990s, most high-income countries had switched to DTaP, although the cheaper DTwP has remained the vaccine of choice in low- and middle-income countries [8]. It is also interesting to note that Poland remains the only country in the European Union that uses wP in the national childhood immunization scheme [7].

Investigations on the possible association of serious adverse reactions with the use of the wP vaccine finally failed to yield any evidence [22,23]. Consequently, the World Health Organization (WHO) published its position that wP vaccines can be used to cover primary doses and first boosters in children under 7 years of age, as there is no evidence to contraindicate their use [24]. Thus, either wP or aP vaccines can be used in the population under 7 years of age. Similar to wP vaccines, paediatric acellular vaccines (aP) can be presented in combination with other vaccines, in addition to the tetanus and diphtheria component, for example, with *Haemophilus influenzae* serotype b (Hib) or inactivated polio (IPV).

The switch towards the use of aP component in many countries contributed to an increase in the acceptance of pertussis vaccinations within the population, leading to an increase in vaccination coverage [9,21,25]. However, a progressive rise in the number of cases has been observed for more than twenty years in Europe, as well as in other regions of the world [26,27,28,29,30,31,32]. In addition, increased incidence of pertussis has been reported in older children, adolescents, and adults [33,34,35,36]. Although the surveillance of the disease in adults is deficient in most countries, the increased incidence of pertussis in this age group has been observed mainly in regions where aP vaccines are administered in early childhood [37]. Accordingly, published reports stated that the immunity induced by aP vaccines wane more rapidly than those induced by wP vaccines [27,28,38]. DTaP vaccination induces an adequate, but not durable, immune response [39]. In fact, the switch from wP to aP in the primary infant immunization was proposed as being at least partially responsible for the increase in cases reported in several countries [40]. As the result, the World Health Organization (WHO) recommended that the switch be considered only if boosters can be assured in the national immunization schedules [24] (*Pertussis Vaccines: WHO Position Paper*, August 2015—recommendations, 2016; https://www.who.int/publications/i/item/WHO-WER9035; accessed 27 November 2022).

The increased incidence of pertussis has triggered a wide-ranging expert debate on the effectiveness of current pertussis prevention strategies. According to expert considerations, the most likely causes for the increased pertussis morbidity rates (Figure 1) include:Evolution of *B. pertussis* under selective pressure exerted by vaccines, particularly aP vaccines [41];Waning of the immunity conferred by vaccination (faster in the case of aP vaccines) [42,43,44];*B. pertussis* transmission by asymptomatic carriers [45];Improvements in pertussis surveillance and in diagnostic methods.

Low vaccination coverage and increased rejection of the use of vaccines are also factors that affect the epidemiology of pertussis. Recently, the anti-vaccine movement was included by the WHO in its report on the ten greatest risks to global health (https://www.who.int/news-room/spotlight/ten-threats-to-global-health-in-2019; Accessed on 27 November 2022). The movement is seen as a threat to the progress made in the fight against vaccine-preventable diseases. The reasons why people choose not to get vaccinated are complex and include lack of confidence, complacency, and access difficulties. There are also those who claim religious reasons for not vaccinating themselves or their children.

In recent years, much attention has been given in the literature to the increased pertussis morbidity rates, in the context of vaccination strategies and the efficacy of available vaccines [46,47]. The current article presents a systematic review of scientific reports on the evaluation of the use of wP and aP vaccines, in the context of long-term immunity and vaccines efficacy.

## 2. Acellular and Whole-Cell Pertussis Vaccine Induce Different Immune Responses

The natural immunity resulting from previous infection is considered more effective than the immunity resulting from vaccinations, and pertussis is no exception [48]. During the recognition stage, *B. pertussis* is detected by the resident innate immune cells within the host, providing the first line of immediate defense [49,50]. Upon the recognition of the bacteria, alveolar macrophages present on the mucosal surface of the lungs phagocytose and kill bacteria rapidly upon their recognition [51]. On the other hand, bacterial virulence factors exert potent immunoregulatory activities that can result in modulation of pathogenesis by suppression of antigen delivery into the draining lymph nodes or enhancing transmission of *B. pertussis*. It was demonstrated that the pertussis toxin (PT) limits the early adaptive immune response by suppressing dendritic cells migration to the draining lymph nodes from the lungs [52]. In addition, Holubova et al. demonstrated that adhesins of *B. pertussis*, the filamentous hemagglutinin (Fha), and fimbriae (Fim) are critical for the infection of nasal cavity and transmission [53]. The difference between vaccination and natural infection, with regard to the triggering of the immune response, resides mainly in the different routes of entry (mucosa vs. systemic) and presentation of antigens/immunogens to antigen-presenting cells. Several authors have also described differences between the immunity induced by aP immunization and that induced by natural infection and wP vaccines [54,55]. It was demonstrated that natural infection and wP vaccination induce Th1- and Th17-dominated responses [56,57,58,59,60]. In contrast, aP vaccines formulated with alum as an adjuvant induce Th2-dominated responses, as evidenced by a significant increase in the levels of cytokines such as IL-4 and IL-5, and a slight increase in IFN-γ levels [61].

Recent scientific reports described the importance of specific tissue-resident memory T cells (Trm) within the upper and lower respiratory tract in both reducing transmission and inducing long-term protection against pertussis [62,63,64]. IL-17A secreted by CD4^+^ Trm cells have been shown to be involved in the clearance of *B. pertussis* from the nasopharynx and to promote the recruitment of neutrophils, particularly Siglec-F^+^ (a lectin normally expressed on mouse eosinophils) neutrophils, to the nasal mucosa upon reinfection [65]. According to a study by Borkner et al. [64], mice lacking IL-17 are incapable of clearing *B. pertussis* from nasal mucosa. In the course of *B. pertussis* infection, CD69^+^CD4^+^ Trm cells producing IL-17 accumulate within the upper and lower respiratory tract, including the nasal cavity, and their levels rise significantly following re-exposure to provide protection against colonization and reinfection [62]. The results presented by Wilk et al. [65] show that, in contrast to immunization with an aP-containing vaccine, immunization with a wP-containing vaccine results in the accumulation of CD69^+^CD4^+^ Trm cells within the upper respiratory tract following re-exposure to *B. pertussis* in mice. Additionally, in baboons, the administration of a wP-containing vaccine was shown to significantly reduce the *B. pertussis* loads within the nasopharynx and prevent the transmission of bacteria to other, non-infected animals [66]. In the same study, immunization with an aP-containing vaccine was shown to protect the animals from the symptoms of disease, while being unable to shorten the *B. pertussis* dwell times within the nasopharynx and prevent transmission to non-infected baboons [66]. These differences may explain why immunization with aP vaccines is not sufficient for inducing local response to infection and counteracting the persistent burden of *B. pertussis* within the upper respiratory tract [66,67]. Moreover, in individuals vaccinated with aP, no increase in Th1 and Th17 lymphocyte activity has been detected, even after the subsequent revaccinations with aP vaccines [38,44,68]. In contrast, the expansion of Th1 and Th17 cells was described in individuals immunized with wP vaccine and boosted with wP or aP vaccines [69]. The main differences in the immune responses induced by aP- and wP-containing vaccines are illustrated in Figure 2.

Available immunological data indicate that Th1 and Th17-mediated immune responses and formation of Trm cells are required for bacterial clearance and long-lasting protection. However, antibodies are equally important providing protection against disease [70]. While the T cell immunity induced by the two types of vaccines has been well-studied, much less attention has been paid to B cell responses following pertussis vaccination, even though the potentially effective pertussis vaccines were selected on the basis of antigen-specific antibody (Ab) levels. In order to identify better markers of vaccine-induced memory, Weaver et al. [70] analyzed long-term immunity from pertussis vaccines and observed that responses to wP vaccination were characterized by a significant increase in T follicular helper cells in the draining lymph nodes (dLNs) and CXCL13 levels in sera, compared to aP-immunized mice. In addition, the study revealed that only immunization with wP vaccine resulted in the generation of *B. pertussis*-specific memory B cells [70]. Recent studies also demonstrated that priming mice with wP vaccines led to faster and stronger stimulation of local germinal centers stimulation and memory B cell responses within the dLNs, as well as to plasma cells induced within the bone marrow presenting with a broader antigenic and isotypic diversity, in comparison to prime immunization with aP vaccine [71]. Importantly, the results indicated that, in contrast to T cell responses, where the first dose of aP vaccine determines the immune response to a boost vaccination with wP, the B cell profile and antibody production induced by aP were re-oriented by a wP boost and led to induction of Ab with a broader isotypic diversity [71]. Whether this re-oriented profile of B cell response is long-lasting or wanes soon after revaccination has not been confirmed. Additionally, high-dimensional flow cytometry was used to characterize the kinetics of B cell subsets circulating in the blood of human subjects of different ages and different priming backgrounds upon administration of an aP booster [72]. Diks et al. showed that the total and IgG1+ plasma cell responses were stronger in subjects primed with wP vaccines than in individuals primed with aP vaccine [72]. Consistently with these data, results showing significantly higher IgG4 levels in children who had received an aP vaccine at primary immunization than those in children who had received a wP vaccine were presented at the World Association for Infectious Diseases and Immune Disorders (WAidid) Congress [68]. IgG4 antibodies are not capable of activating the complement system, and consequently initiating antibody-dependent phagocytosis [73], and because of that, it is critical for the efficacy of a pertussis vaccine to induce a broad repertoire of antibodies where, with an induction of IgG1, antibodies being superior to IgG4 subclass [68]. In Figure 3, we summarize the differences detected in T and B cells, according to the type of vaccine used in the primary schemes and in the boosters. Altogether, it is important to know whether a B cell bias with broader isotypic profile is seen in humans who received a wP booster after the initial aP dose. This type of study will be valuable for implementation of appropriate life-course vaccination strategy in countries switching from the wP to the aP vaccine.

## 3. The Type of Pertussis Vaccine Affects Both the Response in Later Life and the Loss-of-Immunity Rate

Scientific reports indicate that the immunity persists for 10 and 20 years following natural infection and between 10 and 12 years following whole cell vaccination, as compared to about 3 to 5 years following immunization with acellular vaccines [74,75,76,77,78]. The shorter-lasting immunity makes children susceptible to the disease at a younger age (before the administration of subsequent booster doses) and translates to incomplete population immunity. Long-term efficacy studies conducted in Europe and Senegal on children who received 3- or 4-dose series of vaccines suggested that protection waned faster after aP than after wP [79,80,81]. After both three-dose and five-dose primary series of aP vaccination, the protection predictably wanes after the last dose in the series, with the odds of pertussis increasing by a factor of 1.33 (95% CI 1.23–1.43) for every year after administration of diphtheria, tetanus, and acellular pertussis (DTaP) [78]. Epidemiological studies confirm that protection, whereas robustness at the time of vaccination is temporary, with immunity waning as the post-vaccination years pass [76]. The results of the study carried out by Clark et al. [76] show that children who had been fully vaccinated with aP-containing vaccines in infancy were more likely to develop pertussis during the early school years, whereas children who had been vaccinated with wP-containing vaccines were at a higher risk of developing the disease during the adolescence period. Similar differences were also reported by the Vickers’ team [82]. In contrast, in a Swedish 10-year follow-up study, 13% of the population acquired pertussis at a median of 5.5 years after the last dose, irrespective of the type of vaccine (two-component JNIH-6 aP or monovalent Wellcome wP-containing vaccine) [83]. Additionally, in a German 6-year follow-up study of wP and aP vaccines (as manufactured by Wyeth-Lederle), the calculated efficacy for the 6-year follow-up period was 89% (95% CI 79–94) for the aP vaccine and 92% (95% CI 84–96) for the wP vaccine [80]. The limited comparability between studies may result from the use of different vaccines, changes in the manufacturing process or vaccine contents over time, the presence of different immunization schedules (timing and/or number of doses), and the use of different case definitions, surveillance methods and reporting systems.

The decrease in immunity was also reported in those who had completed a full immunization schedule with aP-containing vaccines [84]. In the evaluation of pertussis disease risk in 10- to 17-year-old teenagers, as conducted by Klein et al. [43] after a 2010/2011 pertussis outbreak in the United States, the authors showed that, during the outbreak, DTwP vaccines received in childhood protected the teenagers five times more effectively than DTaP vaccines. Furthermore, the authors estimated the efficacy of DTwP vaccines administered to preschoolers in the United States by investigating the secondary attack rates in household contacts. The results showed that the DTwP vaccine was highly effective in preventing pertussis in preschool children exposed to infection within their households, with protection increasing from 44% for one DTwP vaccine dose to 80% for four or more doses when typical paroxysmal cough was used as a clinical case definition of the disease [85]. Results obtained in the meta-analysis of 11 studies assessing the long-term protection against pertussis after three or five doses of multivalent aP-containing vaccines administered [78] showed that the risk of developing the disease after receiving the last dose of an aP-containing vaccine was estimated to increase by a factor of 1.33 each year (95% CI 1.23–1.43), leading to a conclusion that 8.5 years after the last dose of an aP vaccination schedule, the protection against the disease was maintained only in 10% of children. In another meta-analysis of the efficacy of aP-containing vaccines delivered in accordance with the U.S. immunization schedule [86], the efficacy of the regimen involving administration of six vaccine doses was estimated at 85% (95% CI: 84–86%) in the first year after last dose, with a year-to-year decrease rate of 11.7% (95% CI: 11.1–12.3%), warranting the claim that the immunization program in force might lead to the post-vaccination protection being reduced to 28.2% (95% CI: 27–29%) in patients at the age of 18 [86]. Therefore, it appears that the duration of immune protection following immunization with aP-containing vaccines is insufficient, regardless of the vaccination schedule and the number of doses administered. Increasing the number of aP boosters doses seems to provide short-lasting and rapidly waning protection [44].

In line with what has been described, several studies showed that a combined vaccination schedules that include at least one (at least first) dose of a wP in the vaccination schedule provide a better protection against the disease than those schedules that only consist in the administrations of doses with aP vaccine. Studies showed that the polarization of immune response following the primary immunization in infancy is associated with increased levels of IL-4, IL-9, and TGF-β in children receiving aP-containing vaccines or IFN-γ and IL-17 in children receiving wP-containing vaccines [70,87,88,89,90]. It is considered that the production of IL-9 and IL-17, which varies, particularly depending on the type of anti-pertussis vaccine administered in childhood, may play a significant role in determining this response in later life [69]. Stronger IL-17 polarization from wP vaccination has been reported to be associated with higher protection in baboon models and to be pivotal in the mediation of adaptive immunity by tissue-resident memory T cells after natural infection in mice [39,56].

## 4. Evolution of *B. pertussis*—The Role of Selection Pressure Exerted by Vaccines

An additional hypothesis behind the increase in the incidence of pertussis consists of the pathogen’s adaptation, enabling it to escape vaccine-induced immunity [26]. The first reports regarding the evolution of the *B. pertussis* pathogen were related to the polymorphism in genes encoding for proteins included in the vaccine (such as pertactin [PRN] and pertussis toxin [PTx]) and later in the pertussis toxin promoter (*ptx*P) [90,91]. It has been argued that the changes over time may well be due to natural variation, rather than to immunity-driven selection. Nevertheless, the authors noted that shifts in the *B. pertussis* population coincide with changes in vaccination policy [92,93]. In particular, in countries that only use aP in their calendars, it was reported that PRN-deficient isolates [PRN(-)] increased substantially in the last years [93,94,95]. A study carried out in the United States has shown that the prevalence of pertactin gene-deficient strains could be a consequence of the introduction of aP-containing vaccines in vaccination schedules [93]. Additionally, in Japan, where aP vaccines containing pertactin as one of the antigens had been administered for many years, the frequency of isolates showing no expression of the pertactin gene increased over time from 0% in years 1990–1994 and 0% in years 1995–1999 to 27% in years 2000–2004 and 25% in years 2005–2009 [96]. In Denmark, where a pertactin-free aP vaccine is used, pertactin gene expression is commonly detected in isolates obtained from infected patients [21,97]. Moreover, studies conducted in countries where wP vaccines are used in in the primary vaccination series show a low prevalence of pertactin-deficient strains [98]. It was speculated that this deficiency has less advantage to emerge in a wP-immunized population [99].

The appearance of a wide variety of PRN mutations, each arising from a diversity of *B. pertussis* lineages over time, provides additional strong evidence in favor of vaccine-driven selection on PRN in particular [100]. Longhuan Ma et al. speculated that there are at least three non-mutually exclusive aspects of PRN for its selective loss: its functional redundancy, the relatively longer functional persistence of antibodies against it, and its close location to the surface membrane for productive complement fixation. These characteristics of PRN can be contrasted with PTx, which, for example, requires a complex operon to assemble and export; it has a central and nonredundant role in the pathogenesis of *B. pertussis* and has no paralogs in the genome that can replace it [101,102,103].

Longhuan Ma et al. remark that the lessons learned from this instance should be considered to improve the current pertussis vaccines: “In designing new vaccines, it would be prudent to carefully consider the issues that appear to be enabling potential vaccine escape mutants, such as PRN-deficient strains, to rapidly expand and rise to prominence” [100].

It appears that the dynamics of the selection pressure exerted on *B. pertussis* strains has increased following the widespread introduction of DTaP vaccines, probably due to the reduced variety of antigens in the vaccine composition [91]. In agreement with this hypothesis, a genomic analysis of *B. pertussis* strains carried out in the United Kingdom showed that genes encoding for aP vaccine antigens have been found to evolve at a faster pace than genes encoding for other *B. pertussis* surface proteins [104,105]. Furthermore, a study was recently carried out on more than 3300 strains from 23 countries, which showed that, on average, there are more than 28 transmission chains circulating within a subnational region [106]. This study allowed us to detect that it took 5 to 10 years for *B. pertussis* to distribute homogeneously throughout Europe, with the same period of time required for the United States. The increased fitness of pertactin-deficient strains after implementation of acellular vaccines, but reduced fitness may otherwise explain long-term genotype dynamics [106]. These findings highlight the role of vaccine policy in changing the local diversity of a pathogen that is responsible for 160,000 deaths a year.

## 5. Boosters and the Relevance of Immunization during Pregnancy

Because of waning immunity, boosters are required to sustain pertussis immune response over time. In this sense, and with the objective to protect the most vulnerable population, in 2011, the Advisory Committee on Immunization Practices (ACIP) recommended the administration of aP boosters to all unvaccinated pregnant women, who are one of the main sources of infection in newborns and infants. This aP booster is performed with a pertussis vaccine that contains lower PTX and diphtheria toxoid quantities (Tdap) than the pediatric DTaP vaccine. In fact, the Tdap vaccines are the unique vaccines approved for individuals older than 7 years old. In 2012, the recommendation was updated to address vaccination of all women during the third trimester of pregnancy, regardless of previous Tdap vaccination status (Centers for Disease Control and Prevention, 2013) [107]. Soon afterwards, several ministries/secretaries of health adopted the recommendation, although the strategy has not been universally accepted to date. The strategy has been found to be safe for pregnant women and the developing fetuses and newborns, as well as highly effective in preventing pertussis in infants younger than 3 months, in whom the primary vaccination series has not yet been completed [108,109,110,111].

During a phase I study consisting of a randomized, double-masked, placebo-controlled clinical trial conducted in Houston and Seattle from 2008 to 2012, no Tdap-associated serious adverse events occurred in women or infants [85]. Significantly higher concentrations of pertussis antibodies were measured at delivery in women who received Tdap (n = 33) during pregnancy, as well as in their infants at birth and at 2 months of age, when compared to infants born to postpartum-immunized mothers. In infants born to mothers who had received Tdap during pregnancy, the antibody responses were modestly lower after three DTaP doses, but not different after the fourth dose [112].

A systematic revision published in 2020 showed that immunization of pregnant women with a Tdap vaccine can prevent about 70–90% of pertussis cases and up to 90.5% of pertussis-related hospitalizations in infants under 3 months of age [113]. As different countries incorporate the strategy of vaccination during pregnancy, more evidence is being obtained about its positive effect on the pertussis protection in infants [108]. However, some reports have shown that Tdap immunization in pregnancy is associated with decreases in humoral immune responses to the subsequent immunization of infants using acellular pertussis antigen-containing vaccines. In particular, lower anti-PT IgG levels were detected in infants born to Tdap-vaccinated pregnant women after the completion of primary immunization, while less consistent results were obtained following booster immunization [114,115,116,117]. Data from a meta-analysis of 10 studies has recently shown that anti-*B. pertussis* IgG concentrations in infants/children after primary and booster vaccination were lower for the pertussis antigens included in aP vaccines [118]. Other data from a study carried out in England to evaluate the impact of aP (dTaP3-IPV (diphtheria toxoid, tetanus toxoid, three-antigen acellular pertussis, and inactivated polio), compared with dTaP5-IPV (diphtheria toxoid, tetanus toxoid, five-antigen acellular pertussis, and inactivated polio)), administered during pregnancy on the immunity induced after infant immunization with acellular vaccines, showed no difference in the levels of IgGs against pertussis-specific antigens in children prior to receiving the preschool booster at around 3.3 years of age [119]. The levels of IgGs against the pertussis toxin, however, were significantly lower in vaccinated 3.5-year-olds born to women vaccinated with dTaP3-IPV, as compared to children born to unvaccinated women receiving their vaccination at the same age. Importantly, this effect of antenatal pertussis vaccine on pertussis responses in children is overcome by the administration of a booster dose. It is important to note that none of the participants in the England study were reported to have had confirmed or suspected pertussis disease at any time.

Regarding the immunity induced by wP vaccines administered in infancy to children born to vaccinated women, antenatal Tdap vaccination was reported to inhibit more pertussis-specific responses in wP-vaccinated infants, as compared to aP-vaccinated infants [120]. Antibody functionality, however, was better in the wP groups. In another study, no correlation was found between low anti-*B. pertussis* antibody levels at delivery in infants born to unvaccinated women and their anti-*B. pertussis* antibody levels after wP vaccination [121].

The point that we consider important to highlight here is that, beyond the possible interference of maternal antibodies in the induction of the immune response of the infant doses, surveillance data in countries that had implemented a booster dose against pertussis during pregnancy did not reveal any increase in the number of pertussis cases later in childhood [122]. These data show, at least so far, a possible lack of clinical relevance of prenatal vaccination.

## 6. Discussion

The efficacy of pertussis vaccines has been regularly debated [95,123,124]. Current vaccines, both the traditional vaccine based on a suspension of the heat-killed and detoxified disease causative agent *B. pertussis* and the acellular vaccines made up of purified *B. pertussis* protein immunogens adjuvanted with Alum, have undoubtedly proved to be pivotal for the reduction in the morbidity and mortality associated with the disease. This reduction in the number of cases had been sustained for years; however, a few decades ago, the disease resurged, even in countries with high vaccination coverage [28,125,126,127,128]. The ability of pertussis vaccines to prevent transmission and provide long-term herd protection remains a major point of contention [129]. A large body of evidence presented in this article shows that there are significant differences in the immunity induced by the two different types of vaccines currently in use [56,58,59,65,66,69]. The main difference consisting in the aP-induced Th2 response and the wP-related predominance of Th1 and Th17 responses seems to be of particular importance in shaping the direct protection against pertussis [129]. Considering that pertussis is a disease that most commonly affects young children, and considering the natural tendency of newborns and infants to develop a Th2-type immune response, it appears that the stimulation of the Th1 type at some point in childhood with 1 dose of wP is essential for adequate protection against disease [48]. Data presented in this article also highlight the role of the use of different pertussis vaccine components in providing long-term protection against pertussis. Recent studies have demonstrated that tissue-resident memory T cells play a critical role in maintaining long-term protective immunity to viral and bacterial infections at mucosal surfaces [130]. Using animal models, it was shown that a *B. pertussis* infection induces, within the lungs and nasal tissue, specific CD4 Trm cells presenting with Th1 and Th17 cytokine profiles [62], and these Trm cells play a critical role in maintaining sustained protective immunity against reinfection. The immunization of mice with wP- but not with aP-containing vaccines also primes the IL-17-producing Trm cells. It was proposed that this was the reason why the aP vaccines failed to protect against nasal colonization with *B. pertussis* [66,131]. As shown in the cited reports, while the ability of vaccines containing wP to block the transmission of *B. pertussis* by the nasopharyngeal clearance remains a subject of investigation, the overwhelming majority of reports admit that the whole cell pertussis component-containing vaccines are efficient in providing herd immunity [132,133,134]. On the other hand, the results indicate that aP allow the *B. pertussis* to circulate in the environment, leaving a reservoir of asymptomatic vectors capable of transmitting the infection [45,66].

This, in consequence, causes a risk of the cocoon strategy being ineffective to protect the most vulnerable individuals and makes it necessary to use additional vaccinations in the vaccination schedule. In contrast, the acellular vaccine should be recommended as a booster for women during pregnancy. The data collected since this recommendation was issued in 2011 and shows that the strategy is effective, in terms of the protection induced in newborns, especially during the very early age period, when they cannot receive a vaccine dose [108,113]. Without any doubt, the current knowledge of the induced immune responses and the effectiveness and safety data of the pertussis vaccines being in use today allow for the best use being made of these vaccines in the short-term. However, the scientific community broadly agrees on the need for a third-generation pertussis vaccine [135]. In fact, several vaccine candidates are being evaluated, some in preclinical assays and others in clinical phases [96]. Live attenuated pertussis vaccine, pertussis outer membrane vesicle (OMV) vaccine, or aP vaccine formulated with novel adjuvants have been already shown to be able to induce cellular immune responses in the respiratory tract in animal models, especially when delivered by the intranasal route [38,136,137,138,139,140].

Additionally, results from a phase 2/3 randomized-controlled clinical trial with a monovalent pertussis vaccine containing genetically inactivated pertussis toxin (aPgen) or its combination with tetanus and diphtheria toxoids (TdaPgen) versusa chemically detoxified comparator vaccine (Tdapchem) were recently published (antibody persistence 2 and 3 years after booster vaccination of adolescents with recombinant acellular pertussis monovalent aPgen or combined TdaPgen vaccines) [141].

Three years post-vaccination, the seroconversion rates for PT-neutralizing antibodies were 65.0% (95% CI 44.1–85.9) and 55.0% (95% CI 33.2–76.8) in aPgen and TdaPgen recipients, respectively. Based on these results, the genetically detoxified monovalent aPgen and TdaPgen vaccines can be expected to induce longer-lasting protection than chemically inactivated Tdap vaccines. The safety, colonization, and immunogenicity data for another new, live attenuated pertussis vaccine, BPZE1, have also been published [142]. None of the participants in the clinical trial of this vaccine had presented with any spasmodic cough, difficulties in breathing, or vital signs-related adverse events following immunization. Different doses of BPZE1 have been tested, and all of them were shown to be well-tolerated. Colonization occurring at least once after vaccination was observed in 29 (81%; 68-93) of 36 vaccinated participants. The tested vaccine doses were immunogenic, with increases in serum IgG and IgA titers against the four *B. pertussis* antigens, as compared between baseline and 12 months.

## 7. Conclusions

Research in the area of pertussis vaccinology is focused on defining the minimum criteria that new vaccines must meet in order to be implemented as easily as possible within current vaccination schedules. The most important issues to be considered during the design of a new pertussis vaccine are how to safely induce a specific long-term immune response, how to ensure protection of nasopharyngeal environment to reduce transmission, and how to avoid generating significant selection pressure on the circulating bacterial population. Furthermore, it must be analyzed whether heterologous prime-boost schedule that combine different vaccines during the prime and boost phases that target the same antigen or the prime-pull strategy that relies on two steps: (1) conventional parenteral vaccination to elicit systemic T cell responses (prime), followed by (2) recruitment of activated T cells via topical administration of a T cell attractant (pull), would generate the strongest and long-lasting protection. Finally, novel vaccines based on acellular components should be supplemented with the additional pertussis antigens critical for bacterial transmission and nasopharyngeal mucosa invasion [53]. Vaccines with multiple epitopes, which would induce a memory immune response in the upper and lower respiratory tract with more Th1/Th17 response profile and generate immunity against a wider range of *B. pertussis* strains, seem to be desirable. New adjuvants and different immunization routes could contribute to reaching this objective, which is urgent for better control of the disease. While working in this direction, the strategic use of current vaccines is imperative: primary schedules that include at least some dose of the wP component and aP boosters during pregnancy should be considered.

## Figures and Tables

**Figure 1 vaccines-11-00001-f001:**
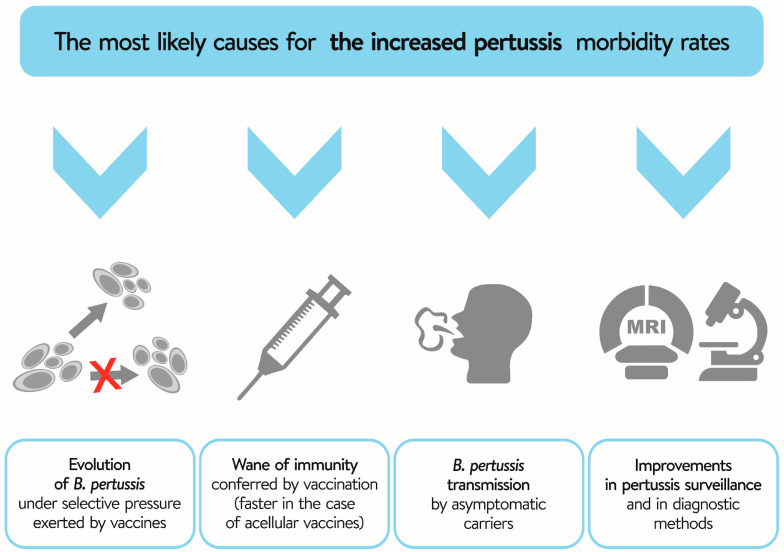
Plausible causes of the increased incidence of whooping cough in the last decades.

**Figure 2 vaccines-11-00001-f002:**
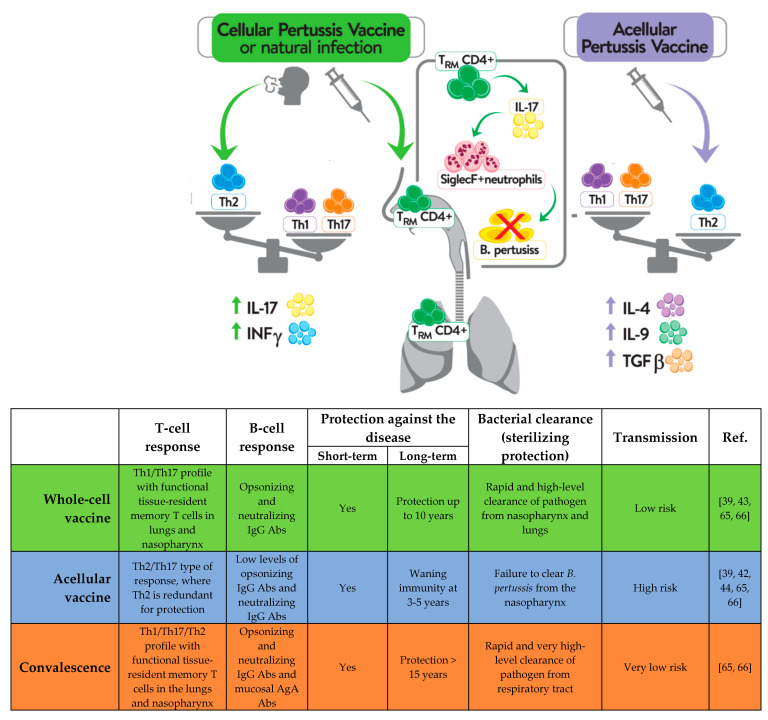
Different immune response to natural *B. pertussis* infection and immunization with vaccines containing a whole-cell pertussis component or an acellular pertussis component.

**Figure 3 vaccines-11-00001-f003:**
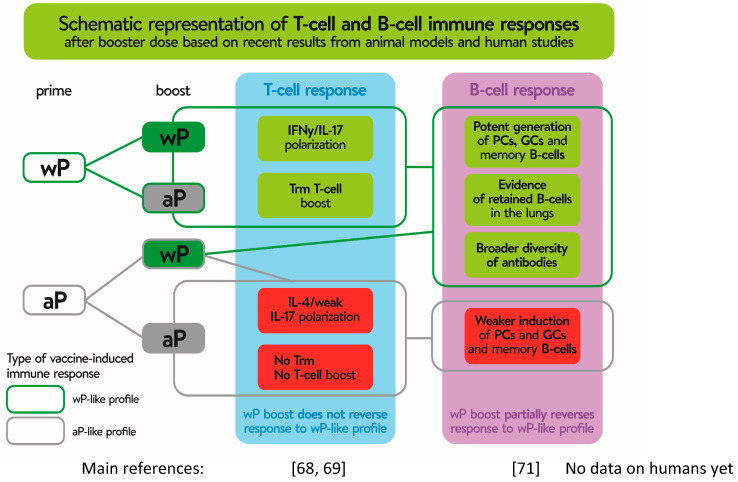
Different T and B cell response, depending on use whole-cell (wP) or acellular (aP) pertussis component during priming and booster immunization. Gray boxes show the effect of priming and booster immunization with aP vaccines. Green boxes show the same effect on T and B cells after prime immunization with wP, irrespective of aP or wP use as a booster, while reversion of the B cell profile after using as a booster wP after priming with aP. Stronger and faster effects of the boost dose on either T or B cell response are shown on a green background. Non-or negative effect on lymphocyte stimulation after boost vaccination is shown in red background. Trm—tissue-resident memory CD4 T cells; PCs—plasma cells; GCs—germinal centers.

## Data Availability

Not applicable.

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
