# Peer review of "Evaluation of Whole-Cell and Acellular Pertussis Vaccines in the Context of Long-Term Herd Immunity"

_vaccines, 2022, doi:10.3390/vaccines11010001_

Round 1
Reviewer 1 Report
Interesting review, relevant topic. The flow is good overall.
Introduction: At the beginning of the introduction, it would be good to include a few lines on the disease and pathogen itself, for context, including susceptibility risks (infants, elderly, asthmatics/COPD patients).
Needs moderate to extensive English editing as there are many grammatical errors throughout the manuscript that should be corrected prior to publication. I would recommend that an English editor or native language speaker with scientific writing experience correct the grammar. This will improve the review. I did not correct the vast majority of grammar mistakes in my comments below.
e.g. Title: Evaluation of whole-cell and acellular pertussis component use in context of long-term herd immunity
Should be: in the context…
Line 36: nowadays vaccines à current vaccines / currently used vaccines
Line 57: The vaccine had an improved safety profile over the Japanese whole-cell
vaccine with comparable efficacy. A reference/reference should be inserted for this line.
Line 61: the abbreviation “aP” is used for acellular pertussis. As this seems to be the first use of the abbreviation, the abbreviation should be explained here.
Lines 65-70: Reference needed. (Other countries adopted similar methods of producing acellular vaccines, usually with additional antigens, such as pertactin and serotype 2 and 3 fimbriae combined with diphtheria, tetanus toxoids (DTaP). Preparations containing up to five components were developed, and several efficacy trials clearly demonstrated that the aP vaccines were able to confer comparable short-term protection than the most effective wP vaccines with fewer local and systemic reactions)
Line 87: Despite the high percentage of vaccinated children, a progressive rise in the number of cases has been observed for more than twenty years in Europe as well as in other regions of the world
[14-20]: While the type of vaccine used may certainly play a role, as outlined in this paragraph, another factor that should be taken into account is the growing number of vaccine deniers. It would be relevant and interesting here to include a graph or table of percentages of vaccinated children over time and the incidence of pertussis over time in various countries, indicating whether aP or wP is used in these countries (or both). This may help to identify the contributions of vaccine type and vaccine denial to the increasing incidence in pertussis. If such a graph already exists in the literature, you could cite it instead and add a brief discussion on it, since it could certainly be a confounding factor.
Line 93: In agreement with this, it has been published that the immunity induced by aP vaccines wane more rapidly than that induced by wP vaccines. Please include the reference here.
Figure 1 title: Here, pertussis it is mentioned as whooping cough for the first time. This information should be in the introduction (see comment above).
In my opinion, the Figure one heading within the figure (blue heading), would look better if the grey box/side flaps behind the blue box are removed and the blue box is instead extended to the appropriate length. The same for the other figures.
Line 108 (and in Figure 1): Evolution of B. pertussis with selection pressure exerted by vaccines. Better (grammatically): Evolution of B. pertussis under selective pressure exerted by vaccines
Line 119: in context of à in the context of
Line 127: …and only different T-cell a slight increase in INF-γ levels
1 Correct abbreviation to IFN-g.
This part of the sentence does not make sense, perhaps something is missing?
Section 2. The difference between immune responses to wP and aP is outlined, and wP induces immune responses more similar to those seen in natural infection. It would additionally be interesting to know: are there any differences between immune responses induced by natural infection and wP? Is it known which components of wP induce the necessary TH1 and Th17 responses? What adjuvants (if any) are used in aP?
Fig 2: Correct abbreviation to IFN-g
In the paper cited, Borkner et al. showed that depleting neutrophils with anti-Ly6G affected the clearance of B. pertussis. As they did not show a specific depletion of SiglecF+ neutrophils, it would be more accurate in the Fig. 2 to write “neutrophils” rather than SiglecF+ neutrophils.
Line 189: It is critical for efficacy of a vaccine to induce a broad repertoire of antibodies where induction of IgG1 antibodies is superior to IgG4 subclass.
1) This may not be true for all pathogens. Perhaps it can be changed to “pertussis vaccine”
2) A reference should be given for this statement, or else, if it from the same Diks et al paper, then this should be made clear
Fig 3: The legend on the bottom left is too small. The PC (plasma cell) and GC (germinal centre B cell) abbreviations should be explained in the text and/or the Figure legend.
Line 225-228 and line 256-263 both discuss reference 58 and mention similar facts; perhaps this can be combined?
Section 3: A table would help to summarize this section.
Section 3 title: The type of pertussis component affects the loss-of-immunity rate.
Perhaps better: The type of pertussis vaccine affects the loss-of immunity rate.
Section 4 would follow Section 2 better than it does Section 3, as it currently jumps from immunology to epidemiology and back to immunology again. The authors could consider re-ordering the sections; perhaps the current Section 3 could come first, then the current Section 2 and 4, or Section 2 and 4 could be combined.
Line 306: calendars à schedules
Line 356: Data of a meta-analysis of 10 studies has recently shown lower anti-B. pertussis IgG concentrations after primary and booster vaccination for pertussis antigens included in aP vaccines [92]. In which population? Babies of vaccinated mothers? This should be clarified.
Line 379: In sum, studies on the impact of vaccination during pregnancy on vaccine-induced
infant immunity, suggest that infants born to Tdap-vaccinated mothers may be at increased risk of pertussis later in life. As the next sentences point out, there does not seem to be an effect on susceptibility to pertussis. Therefore this first sentence should be altered, as there is currently not evidence to suggest that infants born to Tdap-vaccinated mothers may be at increased risk of pertussis, only evidence to suggest that infants born to Tdap-vaccinated mothers have lowered IgG levels against pertussis antigens.
Section 6: A table would again be a nice way to summarize the data.
Line 389 (and line 55/Introduction): are any adjuvants present in the vaccine, or just the antigens?
Line 401: pertussis or whooping cough: These alternate terms should be introduced in the Introduction section already.
Lines 410-417: repeats themes of section 2 and should rather be incorporated into this section; perhaps summarized briefly in the Discussion in a sentence.
Discussion:
Asthmatics are also susceptible to pertussis, and should be mentioned as another susceptible group. Are there any studies looking at vaccination efficacy/immune responses to the different vaccines in asthmatics? The elderly are also more susceptible – what is known with regard to them?
If new pertussis vaccines have been preclinically or clinically tested, it would be worth including a short paragraph on this.
Reviewer 2 Report
This manuscript reviewed the pros and cons of inactivated whole cell (wP) vs. acellular pertussis (aP) vaccine, including the likely explanation for the resurgence of whooping cough after switching from wP to aP vaccine. The authors speculated the resurgence was due to microbial evolution, the differential induction of Th1 vs Th2 immunity by wP or aP vacccine, the duration of immunity, and the effect of priming with inactivated whole cell vaccine followed by acellular or component vaccine. The discussion included boosters during pregnancy and the subsequent effect on the quality and duration of immunity of the child against whooping cough.
A major flaw of this manuscript is in the title: There was no solid data or evidence to show the “long term herd immunity”, there was no “evaluation”, and no data, on the two forms of pertussis vaccine in this regard. Furthermore, the text was written as mere reiteration of the literature rather than a true critical evaluation.
There was also a lack of main theme or thesis. Is inactivated whole cell vaccine better? Should it be re-introduced? If acellular vaccine is inadequate, what about increasing the immunity by more boosters?
The authors should focus on the differences between “sterile immunity” vs. “protective immunity”, not just meandering between inactivated whole cell and acellular vaccine. The layout of the text was convoluted and difficult to follow. There were no citation, and no data, to substantiate the three figures (Fig. 1 to Fig. 3).
The authors should include a summary table to highlight the pros and cons, “protective” vs. “sterile immunity” by wP and aP vaccines, recent epidemiological data on the “resurgence” of whooping cough, and their hypothesis on the duration of immunity, and hence suggestive of [reduced] “herd immunity”.
In summary, this “review” manuscript has no “value-added” to what is known in this field. It should be revised to include the above suggestions.
Reviewer 3 Report
The authors review the literature on the differences in the immune response to whole cell and acellular pertussis vaccines as a likely explanation for the resurgence of pertussis. The concepts that the resurgence is caused by the nature and rapid waning of the immune response to DTaP vaccines and possible emergence of pertactin-negative strains of B. pertussis has been discussed extensively before. The authors discuss recent studies that further support these concepts.
Comments:
1. Line 17 – 20. “Tough” is probably meant to be “Though”. However, this implies a contradiction, but the switch to acellular pertussis vaccines is not inconsistent with, but rather directly related to, an increase in pertussis as discussed in the manuscript. Please rephrase.
2. Line 17. “industrialized” is an outdated term to designate high income countries.
3. Line 24. “were proposed.” Replace by “have been proposed”.
4. Line 24. “are” replace by “is”.
5. Line 36. “nowadays” is not an adjective. Replace by “current” or something similar.
6. Line 38. “to become DTP vaccine, which was widely adopted”. Change to: which was widely adopted as a DTP vaccine.
7. Line 82. Italicize Haemophilus influenzae.
8. Line 83. Insert “the” before European.
9. Line 85-87. The lower rate of adverse effects is not directly related to the increase of pertussis. Please rephrase.
10. Line 95. Replace “wane” by “wanes” .
11. Line 96. “In fact,…..” This is a direct follow-up to the previous sentence and should not be a new paragraph.
12. Line 127. “and only different T-cell a slight increase in INF-γ levels”. This is not clear. I suggest removing “only different T-cell”.
13. Line 132. What is the significance of Siglec-F+ neutrophils (as opposed to other neutrophils)?
14. Line 147 – 150. This is repetitive with line 124 – 127. Delete or rephrase.
15. Line 154 – 159. This is unclear. The immune responses to DTwP and DTaP do not persist. In fact, the argument is that the immune responses to DTaP in particular wane rapidly. The authors should rephrase this to make clear that the type of immune response induced by priming with the DTwP or DTaP is not changed by boosting with aP or wP.
16. Figure 2. The balances go the wrong way! The law of gravity indicates that more Th1/Th17 cells means that the balance should be lower relative to Th2 cells (left side) and vice versa.
17. Line 165. “antibodies are equally important providing protection”. How is this related to the antibody isotype discussed on the following page?
18. Line 175. Replace “lead” by “led” (past tense).
19. Line 193 – 198. “the results presented at the 2018 World Association for Infectious Diseases and Immunological Disorders (WAidid) Congress”. Where these data were presented is not relevant. The reference [48] is also wrong. This work was published in Vaccine (https://doi.org/10.1016/j.vaccine.2017.11.066).
20. Line 198 – Delete “Nevertheless”.
21. Line 214. This heading is not clear. Suggest: “The type of pertussis vaccine affects the duration of immunity.”
22. Line 217: “the wane of the immunity induced…”. Replace by “the waning of immunity following...”.
23. Line 315 – 317. Explain and elaborate – do the whole cell pertussis not contain pertactin?
24. Line 318 – 323. The authors should elaborate and explain in more detail why they believe that the presence of fewer antigens in the aP vaccine would lead to increased selection pressure versus the presence of many antigens contained in wP vaccines.
25. Line 375. How was “antibody functionality” determined? How is this related to the antibody isotype such as IgG4 that was discussed earlier in the manuscript?
26. Line 412 – 415. Change “animal model” to “animal models”. The effect of wP on nasal colonization was also demonstrated by Soumana et al. (2021) (DOI: https://doi.org/10.3201/eid2708.203566).
27. Line 419. Why does this “remain a subject of debate”? Is there evidence to the contrary?
28. Line 438. What do the authors mean by “heterologous prime-boost schedule and prime-pull strategy”?
29. References. Please check the accuracy of the references. The titles of Refs. 41 and 42 seem incomplete.
Round 2
Reviewer 2 Report
This version has signifcantly improved. However, a table to summarize the pros and cons and the nature of immunity (sterile vs. protective; and their duration) is not provided.
Concurring with Reviewer #3, Fig. 2 is problematic: Authors intended to show there was an increase in Th1Th17 by wP vaccine or by natural infection, however, the use of a "balance" symbol suggested that Th1Th17 were decreased (being "lighter"). Likewise for the aP vaccine. Need to change.
